# Flame Retardancy of High-Density Polyethylene Composites with P,N-Doped Cellulose Fibrils

**DOI:** 10.3390/polym12020336

**Published:** 2020-02-05

**Authors:** Shuai Zhang, He Chen, Yin Zhang, Yi-meng Zhang, Weiyan Kan, Mingzhu Pan

**Affiliations:** College of Materials Science and Engineering, Nanjing Forestry University, Nanjing 210037, China; zhangshuai1994@126.com (S.Z.); 13770711109@163.com (H.C.); zhangyin@njfu.edu.cn (Y.Z.); zhangym@njfu.edu.cn (Y.-m.Z.); kanweiyan@163.com (W.K.)

**Keywords:** polymer-matrix composites (PMCs), flame retardancy, microstructural analysis, thermal stability

## Abstract

To derive P,N-doped cellulose fibrils, phosphoric acid and aqueous ammonia were placed in a one-pot reaction, and the phosphate groups and ammonium phosphates were successfully introduced into the cellulose surface. The obtained P,N-doped cellulose fibrils with high liberation were thereafter incorporated into a high-density polyethylene (HDPE) matrix to improve the flame retardancy of HDPE composites, and they had a significant improvement on flame retardancy of HDPE composites. In particular, 7 wt % P,N-doped cellulose fibrils considerably reduced the average and peak heat release rate (HRR) by 29.6% and 72.9%, respectively, and increased the limited oxygen index (LOI) by 30.5%. The presence of phosphate groups and ammonium phosphates within P,N-doped cellulose fibrils was found to promote the thermal degradation of HDPE composites at a lower temperature (i.e., 240 °C). The released acid catalyzed the dehydration of cellulose to form an aromatic carbonaceous structure with a higher crystalline orientation, which improves the flame retardancy of HDPE composites.

## 1. Introduction

Natural fiber reinforced polymer composites (NFRPCs) have been widely applied in fields of decking, railing, construction and packaging because they are biodegradable, renewable, and have excellent mechanical properties [1]. These materials, however, are highly flammable, thus limiting their applications in building, transportation, and furniture manufacturing industries. Phosphorus/nitrogen-based mixtures, such as ammonium polyphosphate, are the main flame retardant additives commonly used in NFRPCs because they have a highly efficient flame retardancy and do not cause environmental problems [2,3,4]. A higher load of phosphorus/nitrogen-based mixtures, however, always leads to higher cost, heterogeneous distribution, and poor compatibility between polymers and composite materials. Attempts have been made to simultaneously improve the flame retardant performance and mechanical properties through the incorporation of nanomaterials, such as nanodiamond [5] and nano-SiO_2_ [6].

Cellulose is the most abundant natural renewable organic polymer found all over the world. The global biomass production, most of which is cellulose, is estimated at approximately 150–200 billion metric tons a year [7]. Recently, the introduction of cellulose with micro/nano scale to polymers as reinforcement has attracted attention because of its excellent mechanical properties and interesting char-forming characteristic. Flame retardants containing P and N elements like chitosan, phytic acid, APP, DOPO are the most commonly appeared elements in environment friendly flame retardants and they are often introduced to NFRPCs. Gaan et al. [8] studied the effects of three nitrogen additives on the flame retardancy of cotton cellulose treated with tributyl phosphate (TBP). Results proved the synergistic effect between phosphorus and nitrogen that brings a better flame retardancy. Shi et al. [9] alternatively deposited PEI and APP on carbon fibers for flame retardancy of epoxy resin. With only 6 bilayers, a high flame retardancy (LOI of 41.0% and UL-94 V0 rating) was achieved. Wang et al. [10] constructed a phosphorus-nitrogen containing polymer wrapped carbon nanotubes (CNT-PD) to improve the flame retardancy of epoxy resins. With 4 wt % addition of CNT-PD, LOI value of epoxy resins reached to 33.6%. Costes et al. [11] investigated cellulose-phosphorus combinations for sustainable flame retardant polylactide (PLA). Polylactide composites that contain 20 wt % nanocrystalline cellulose (NCC) or 10 wt % NCC/aluminum-phytate (Al-Phyt) significantly reduce the peak heat release rate (HRR) from 390 kW/m^2^ (for neat PLA) to 255 (20 wt % NCC) and 240 kW/m^2^ (10 wt % NCC/Al-Phyt), respectively. Li et al. [12] prepared the NCC with phosphoric acid hydrolysis, and used it to modify polyurethane foam (PUF). They reported that PUF with 6 wt % NCC has a low HRR of 49 kW/m^2^ and a high compressive strength of 107.5 kPa. Ghanadpour et al. [13] fabricated phosphorylated cellulose nanofibrils (CNF) using (NH_4_)_2_HPO_4_, and utilized them for nanopaper sheet. Phosphate groups that are present in the CNF structure endowed the nanopaper sheets with an improved flame retardancy. Feng et al. [14] fabricated a novel core-shell nanofibrous flame retardant system (PN-FR@CNF) by chemically grafting the phosphorus-nitrogen-based polymer onto the CNF surface in situ. The addition of 10 wt % PN-FR@CNF enables the PLA to achieve a V-0 flame resistance rating and an increased tensile strength. Subsequently, Yin et al. [15] successfully designed a new hybridized flame-retardant system (APP@CNF) by facile ball-milling. The results show that the improved dispersion of APP@CNF enhances the flame retardancy and mechanical strength of PLA composite. Considering the successful application of treated cellulose fibrils as synergistic reinforcement phase and flame retardant in polymers like PLA and PU, its application in one of the most widely used synthetic plastics-polyethylene is still lack of investigation. Though natural cellulose reinforced polyethylene [16,17,18] has been widely studied, flame retardant treatment of such materials still follow a bulk mode that directly adding FRs into the polyethylene matrix and, thus, a high loading of FRs is still required to maintain a satisfactory fire proof level.

Usually, cellulose with micro/nano scale is usually isolated from natural fiber, indicating that there are strong interactions between cellulose, hemicellulose, and lignin [19]. Enormous efforts have been made to utilize sulfuric acid and hydrochloric acid for the preparation of micro/nano cellulose fibrils [20,21]. After acid hydrolysis, the treatments for residual acid catalysts pose significant environmental risks. Furthermore, sulfuric acid hydrolysis can compromise the thermal stability of cellulose due to the grafting of anionic sulfate ester groups on the cellulose surface, inhabiting the potential application of cellulose in related polymer-based composites [22,23]. Among the foregoing mineral acids, phosphoric acid is considered as an alternative to considerably liberate cellulose fibrils and enhance their thermal stability [24,25]. The residual phosphoric acid after acid hydrolysis is among the main components of flame retardant additives that do not require further post-treatments. Hence, the combination of phosphoric acid derived by biomass hydrolysis and flame retardant polymer composites promotes the utilization of cellulose fibrils and affords a means to protect the environment. In our experiment, phosphoric acid and aqueous ammonia were chosen to fabricate P,N-doped coatings mainly for their economic acceptability.

Herein, cellulose is treated with phosphoric acid and aqueous ammonia in a one-pot reaction to produce P,N-doped cellulose fibrils. These cellulose fibrils are subsequently introduced to high-density polyethylene (HDPE) to improve flame retardancy. The formation of P,N-doped cellulose fibril and its influence on thermal degradation, flammability properties, and mechanism of the resultant composites are systematically examined in this study.

## 2. Materials and Methods

### 2.1. Raw Materials

HDPE (5000S) pellets with a melt flow index of 0.8–1.2 g/10 min (190 °C/2.16 kg) and a density of 0.95 g/cm^3^, was kindly provided by Sinopec Yangzi Petrochemical Co., Ltd. (Nanjing, Jiangsu, China). The pellets were ground to a homogeneous powder with a plastic mill (NF-200, Kim Jung-Teaching Instrument Co., Ltd. Nanjing, Jiangsu, China). Rice straw was obtained from the rural area of Nanjing City. The samples were crushed and sieved to obtain granules between 40 and 60 mesh sizes and, thereafter dried at 103 °C for 6 h. Phosphoric acid and aqueous ammonia were purchased from Sinopharm Chemical Reagent Co., Ltd. (Nanjing, Jiangsu, China).

### 2.2. Fabrication of P,N-Doped Cellulose Fibrils

Purified cellulose fiber from rice straw was prepared according to the procedure in our previous paper [26]. Briefly, the rice straw granules were consecutively with benzyl-alcohol mixtures and then delignated following with acetic acid and sodium chlorite. The purified cellulose fiber was subsequently placed in 85 wt % phosphoric acid with a 1:5 ratio (based on the solid mass of cellulose) and stirred for 2.5 h at 50 °C according to the report of Li et al. [12]. The mixtures were then treated with aqueous ammonia in an ice water bath until the system’s pH value reached 7.5. Cellulose fibrils doped with the elementals of phosphorus and nitrogen (named as P,N-doped cellulose fibrils), were subsequently used to reinforce the HDPE. Scheme 1 illustrated the possible reactions during hydrolysis and neutralization with aqueous ammonia.

### 2.3. HDPE Composite Preparation

The HDPE powders were added to P,N-doped cellulose fibril suspension with mechanical stirring. The blends were further vacuum-dried at 60 °C for approximately 24 h. The mixtures were thereafter melt-compounded using a ZG-160 open mill (Dongguan ZhengXin Electromechanical Science and Technology Ltd., Dongguan, Guangdong, China) for 5–10 min at 170 °C. Finally, the blends were compression-molded for 1 min at 175 °C to produce specimens for various measurements. The HDPE composites with P,N-doped cellulose fibrils loadings of 1, 3, 5, 7, and 9 wt % (based on the HDPE solid mass) were designated as P,N-doped 1 wt %, P,N-doped 3 wt %, P,N-doped 5 wt %, P,N-doped 7 wt %, and P,N-doped 9 wt %, respectively. For comparison, the HDPE was also reinforced with 7 wt % of untreated cellulose, named as untreated 7 wt %.

### 2.4. Characterization

Scanning electron microscopy (SEM) measurement was performed using QUANTA 200 instrument (FEI Co., Hillsboro, OR, USA) with an energy dispersive spectroscopy instrument (EDS, OXFORD Instrument, Oxford, UK). The surface of cellulose fibrils and char residues was sprayed with gold using a sputter coater before analysis. The elemental mapping of carbon (C), oxygen (O), phosphorus (P), and nitrogen (N) within samples was also performed.

The Fourier transform infrared (FTIR) spectra of cellulose fibrils and char residues were recorded on a VERTEX 80 infrared spectrum instrument (Bruker, Saarbrücken, Germany) with a 0.5 cm^-1^ resolution with ATR mode.

The X-ray photoelectron spectroscopy (XPS) was conducted to analyze the elements content and distribution within the samples, which were first freeze-dried. A small quantity of samples was thereafter mounted using double-sided adhesive tape, and the XPS spectra were recorded by AXIS UltraDLD (Kratos Co., Kyoto, Japan) with a monochromatic Al Kα source (1486.6 eV), and subsequently analyzed using XPSPEAK41 software. A Shirley ‘baseline’ was used for background subtraction, whereas Gaussian (80%)-Lorentzian (20%) peaks were employed for spectral deconvolution.

The X-ray diffraction (XRD) patterns of cellulose fibrils and char residues were recorded with an Ultima IV diffractometer (Rigaku, Tokyo, Japan) using Cu K_α_ radiation (λ = 0.15406 nm), operating at 40 kV and 30 mA.

The Raman spectra of char residues were obtained by DXR532 Raman spectrometer (Themor, Waltham, Massachusetts, USA) with a 532 nm solid laser as excitation source.

Thermogravimetric analysis (TGA) was performed by utilizing a NETZSCH TG 209 F3 in a nitrogen atmosphere. For each experiment, 1–2 mg of samples were heated at a rate of 10 °C/min within a temperature range of 30–700 °C. The TG-FTIR of cellulose fibril was further performed according to a previous procedure reported by Chen et al. [27]. It was implemented using a TGA Q500 thermal analyzer (TA Instrument, New Castle, Delaware, USA) coupled with a Nicolet 6700 FTIR spectrometer (Thermor, Waltham, MA, USA). Approximately 15 mg of samples were analyzed in a nitrogen atmosphere at a heating rate of 20 °C/min within a temperature range of 30–750 °C. The FTIR spectra were collected in the range 4000–500 cm^−1^ with a resolution of 1 cm^−1^.

The flammability of HDPE composites was determined by cone calorimetry (Fire Testing Technology, East Grinstead, UK) according to ISO 5660 standard. Square samples with dimensions of 100 mm × 100 mm × 4 mm were horizontally exposed to an external heat flux of 50 kW/m^2^. The flammability results, including the HRR, total heat release (THR), time to ignition (TTI), fire growth rate (FGR), effective heat of combustion (EHC), specific extinction area (SEA), total smoke release (TSR), and mass loss rate (MLR), were recorded as the average values of duplicate measurements. Char residues from cone calorimeter were collected to investigate the physical and morphological structures. The limited oxygen index (LOI) test was also conducted to evaluate the fire performance of samples on an HC-2C oxygen index meter (Nanjing Jiangning Analytical Instrument Co., Jiangning, China) according to ISO 4589. The sample dimensions were 100 mm × 10 mm × 4 mm.

## 3. Results

### 3.1. Morphology of P,N-Doped Cellulose Fibrils

The SEM images of untreated and treated cellulose fibrils are shown in Figure 1, and the elemental mapping results of corresponding cellulose fibrils are shown in Figure 2. The microscopic surface of untreated cellulose exhibits partial fibrillation after the removal of hemicellulose and lignin (Figure 1a). It is evident that the C and O elements are homogeneously dispersed throughout the cellulosic fibers (Figure 2a). After phosphoric acid hydrolysis, the fibers pile up on each other with an obvious fibrillation (Figure 1b). The mass percentage of P is significantly increased to 28.16 wt %, whereas the mass percentage of C is barely detected probably due to a coating of excessive phosphoric acid (Figure 2b). After neutralization with ammonia, quantities of highly fibrillated cellulose fibrils are further exposed as a result of a removal of phosphoric acid coating (Figure 1c). Interestingly, the mass percentage of C was significantly increased to 14.60 wt % from 0.12 wt % (P-doped cellulose). Moreover, compared to P-doped cellulose, a decreased content of 16.99 wt % (P elemental) and an increased content of 13.83 wt % (N elemental) is also present in P,N-doped cellulose fibril (Figure 2c).

### 3.2. Chemical Property of P,N-Doped Cellulose Fibrils

Figure 3a shows the FTIR spectra of untreated and treated cellulose fibrils. The spectrum of untreated cellulose exhibits characteristic bands at 3334 cm^−1^ for the stretching vibration of hydroxyl, 2892 cm^−1^ for the symmetric vibration of C-H within the methylene, and stretching vibrations at 1030 cm^-1^ for the C-O-C groups of β-d-glycosidic bonds [11,13]. After the phosphoric acid hydrolysis, the stretching vibration of hydroxyl in the P-doped cellulose is shifted to a lower frequency of 3210 cm^−1^. It indicates the occurrence of reaction between phosphoric acid and cellulose fiber that consequently, forms strong hydrogen bonds. The strong absorption peak at 956 cm^−1^ and minor peak at 885 cm^−1^ correspond to the P-O-C stretching mode and P-O-C aliphatic bond, respectively. The bending vibration of methylene present at 1435 cm^−1^ is also enhanced after phosphoric acid treatment. This indicates an esterification reaction between the alcoholic hydroxyls of cellulose and phosphoric acid, leading to the formation of cellulose phosphate [25,28]. After the neutralization with ammonia, the stretching vibration at bands of 3338–3239 cm^−1^ present in the P,N-doped cellulose exhibits a combination of hydroxyl and amine group. Additionally, the band intensity at 1638 cm^−1^ (–C=O stretching) is enhanced in P-doped and P,N-doped cellulose fibrils.

Figure 3b shows the XRD profile of untreated and treated cellulose. The untreated cellulose shows a characteristic crystal structure of cellulose *I,* that appears at 15.9°, 22.3° and 33.7° according to Miller indices of [110], [002], and [400], respectively [24,29]. As for P,N-doped cellulose, the characteristic diffraction peaks that occur at 17.10°, 24.16°, and 29.44° can be assigned to crystalline planes of [101], [200], and [112] in the crystal structure of (NH_4_)H_2_PO_4_. A crystalline plane of [200] in the crystal structure of (NH_4_)_2_HPO_4_ at a 2*θ* values of 17.97° also appears [30]. It reveals that excessive phosphoric acid remained in P,N-doped cellulose surface could react with ammonia, leading to a formation of crystallized inorganic ammonium phosphates. These ammonium phosphates are formed during the neutralization process, supporting the distribution of C, P, and N elements from SEM-EDS analysis. Additionally, the diffraction peaks of cellulose are rarely found due to the strong diffraction peak of ammonium phosphates.

The XPS can provide information pertaining to the elemental composition and content of untreated and treated cellulose that supplements the FTIR and SEM-EDS results. Figure 3c further shows the XPS spectra of samples, and the peaks at 133, 285, 401, and 531 eV could be assigned to P2p, C1s, N1s, and O1s of untreated and treated cellulose, respectively [31]. As for the P-doped cellulose, the C element is reduced to 28.70 wt % from 53.12 wt % (untreated cellulose), and the P element is significantly increased to 22.51 wt %. As for the P,N-doped cellulose, 17.33 wt % of P and 7.94 wt % of N are obtained apart from C and O. Interestingly, the atomic contents of C, O, P, and N elements in untreated and treated cellulose fibrils remain in the same trend as that indicated by SEM-EDS results. To further demonstrate the structure of treated cellulose, high-resolution scans of the P2p spectra are also presented in Figure 3d,e. The treated cellulose both displays strong peaks at 134 and 135 eV, which can be attributed to the P-O-C or -PO_3_^2-^ groups [13,32]. Noticeably, the P-doped cellulose exhibits a high content of -PO_3_^2-^ groups and a low content of P-O-C probably due to an existence of unreacted phosphoric acid within treated cellulose surface. After neutralization with aqueous ammonia, the P,N-doped cellulose has a relatively high content of P-O-C. An excessive phosphoric acid reacted with ammonia, leading to a formation of ammonium phosphates and an expose of P-O-C within treated cellulose surface. The foregoing is also consistent with the FTIR and SEM-EDS results.

### 3.3. Thermal Property

#### 3.3.1. Thermal Property of P,N-Doped Cellulose Fibrils

Figure 4 shows the TG-FTIR results of untreated and treated cellulose exposed to nitrogen atmosphere. Figure 4a shows that the untreated cellulose exhibits a peak thermal degradation temperature (*T*_peak_) at 360 °C, whereas for the P-doped cellulose, *T*_peak_ is present at a lower temperature of 182 °C. After neutralization with ammonia, *T*_peak_ appears at 260 °C for the P,N-doped cellulose. During the whole thermal degradation, the P,N-doped cellulose shows the highest char residue content, up to 65.04 wt % at 260 °C. This char residue can afford an effective physical barrier effect during the combustion. For the gaseous mixture composition of untreated cellulose fiber (Figure 4b), the peak at 3400 cm^−1^ represents the stretching vibration of hydroxyl. Absorbance bands appear at 2250–2400 cm^−1^ and 2000–2250 cm^−1^ that are assigned to CO_2_ and CO, respectively, although more CO is generated than CO_2_ [33]. For the P-doped cellulose (Figure 4c), the absorption of hydroxyls is strengthened, and the carbonyl group (C=O) at 1739 cm^−1^ is significantly decreased. This indicates that the P-doped cellulose releases fewer flammable volatiles, thus limiting burning [34,35]. After neutralization with ammonia (Figure 4d), the peaks at 3400–3200 cm^−1^ and 1436 cm^−1^ are assigned to the combination of H_2_O and NH_3_. Their intensities are also increased, improving the flame retardancy of HDPE composites. The release of NH_3_ further indicates the presence of ammonium phosphates on the surface of cellulose fibrils.

#### 3.3.2. Thermal Property of HDPE Composites

Figure 5 shows the TG results of HDPE composites with the untreated and P,N-doped cellulose exposed to nitrogen atmosphere. For HDPE composites with 7 wt % untreated cellulose, *T*_5%_ (temperature corresponding to a 5 wt % mass loss) and *T*_peak_ appear at approximately 338 and 475 °C, respectively. For HDPE composites with 7 wt % P,N-doped cellulose, thermal degradation starts at a lower temperature of 240 °C. Its char residues are 13.8 wt % at 700 °C, indicating a significant increase in char formation. Based on the above TG-FTIR analysis of P,N-doped cellulose fibrils, it may be concluded that the formation of cellulose phosphate and ammonium phosphates during phosphoric acid hydrolysis and neutralization can catalyze the dehydration and esterification of cellulose fibrils, promoting the thermal degradation of HDPE composites and higher content of char formation [11,35,36].

### 3.4. Flame Retardancy of HDPE Composites

Table 1 summarizes the flammability properties of HDPE composites with untreated and P,N-doped cellulose fibrils. The HDPE with untreated cellulose has a higher LOI of 20.3% compared to that of neat HDPE. For HDPE with P,N-doped cellulose fibrils, the LOI further increases from 20.4% to 25.7%. In terms of LOI, the HDPE composites with 7 wt % P,N-doped cellulose fibrils are the most effective.

#### MLR-Mass Loss Rate

Figure 6a shows the HRR of HDPE composites reinforced with untreated and P,N-doped cellulose. For the neat HDPE, the maximum HRR is 1519.21 kW/m^2^. The peak HRR of HDPE with 7 wt % untreated cellulose sharply decreases to 1035.73 kW/m^2^, which is approximately 32.8% lower than that of neat HDPE. The HDPE with P,N-doped cellulose fibrils affords better flame retardancy. The most remarkable reduction occurs in the use of HDPE with 7 wt % P,N-doped cellulose: the average and peak HRRs are 185.76, and 412.30 kW/m^2^, which are approximately 29.6% and 72.9% lower than those of neat HDPE, respectively. Furthermore, it is evident that the incorporation of 3–9 wt % P,N-doped cellulose could lower the THR by 8.6%–26.6% (Figure 6b). The FGR values are significantly decreased from 7.58 kW/(m^2^·s) for neat HDPE to 4.34 and 3.10 kW/(m^2^·s) for HDPE composites with 7 and 9 wt % P,N-doped cellulose, respectively. These reductions indicate a low rate of fire spread [37]. The total smoke release (TSR) of HDPE composites is summarized in Table 1 and shown in Figure 6c,). These indicate that the TSR of HDPE with P,N-doped cellulose ranges from 2313.94 (1 wt % P,N-doped cellulose) to 2981.84 m^2^/m^2^ (7 wt % P,N-doped cellulose), which is considerably higher than that of HDPE with 7 wt % untreated cellulose. NFRPCs is often treated with phosphorus flame retardant (APP, red phosphorus), along with the improvement of flame retardancy, phosphorus flame retardant brings the problem of increased smoke release, especially in NFRPCs [38,39]. During the combustion process, phosphorus flame retardants, which act by flame poisoning/gas phase activity, interfere with the typical combustion radical reactions in the flame and result in incomplete combustion and thus brings a higher fire proof level together with a higher TSR index than that of HDPE with 7 wt % untreated cellulose. The incorporation of P,N-doped cellulose increases the char residue of HDPE and suppresses the combustion of HDPE but causes the incomplete burning of volatile matter, thus increasing the TSR of HDPE composites [15]. A strategy for the simultaneous suppression of heat and smoke release to effectively improve the flame retardancy of HDPE composites will be reported in our future work. The LOI and cone calorimeter results show that P,N-doped cellulose fibrils could improve the flame retardancy of HDPE composites. In particular, the highest effectiveness is exhibited by HDPE composites with 7 wt % P,N-doped cellulose.

### 3.5. Morphology and Structure of Char Residues

Figure 7a,b shows the char residue morphology of HDPE composites after combustion. The char residues of HDPE reinforced with 7 wt % untreated cellulose is considerably low in amount and thin (Figure 7a). On the other hand, the HDPE reinforced with 7 wt % P,N-doped cellulose exhibits a thick char residue layer with a few detectable cracks (Figure 7b); this layer, is mainly responsible for the optimum flame retardancy.

To further investigate the microscopic and physical structure of char layer, the char residues of HDPE composites with 7 wt % untreated and P,N-doped cellulose after combustion are examined using SEM-EDS, FTIR, XRD, and Raman measurements.

The microcosmic images of char residues are shown in Figure 7a’,b’. For HDPE composites with 7 wt % untreated cellulose, the carbon framework of char was remained after combustion (Figure 7a’). For HDPE composites with 7 wt % P,N-doped cellulose, the decomposed substance from ammonium phosphates is homogeneously stacked on the carbon framework surface (Figure 7b’), resulting in the formation of a more compact physical structure. From the elemental mapping of char residues (Figure 8), the mass percentage of P and N increases to 25.09 and 8.28 wt %, respectively, in the char residues of HDPE composites with 7 wt % P,N-doped cellulose. The mass percentage of C, however, is significantly decreased to 1.66 wt %, mainly due to the coating of P,N substrates.

The FTIR spectra of char residues of HDPE composites are shown in Figure 9a. An aromatic C=C stretching vibration at 1637 cm^−1^ [40] reveals the network structure of char produced during the combustion. The P=O and P-O-P stretching vibration at 1165 and 995 cm^−1^ indicate a formation of pyrophosphoric acid, respectively [41]. The XRD spectra of combustion char of HDPE composites are shown in Figure 9b. The diffraction peaks at ~24° correspond to crystalline planes with [002] of char from the rapid pyrolysis of cellulose during combustion [42,43]. The more shaped and stronger peak of the char residue of HDPE composites with 7 wt % P,N-doped cellulose fibrils indicates a higher crystallite structure orientation. Figure 9c further shows the Raman spectra of char residue of HDPE composites. Generally, the G-band is attributed to the vibration of in-plane sp^2^-hybridized carbon atoms in the graphite layer, whereas the D-band is related to the vibration of carbon atoms with a dangling band in the plane terminations of disordered graphite or glass carbons, representing the sp^3^-hybridized carbon and the presence of defect-like amorphous domain [44,45]. Two visible bands corresponding to G-band (1587 and 1618 cm^−1^) and D-band (1364 and 1362 cm^−1^) reveal a combination of graphic and disordered carbon structures within the char residue. For HDPE with 7 wt % P,N-doped cellulose, the char residue exhibits a higher G-band (1618 cm^−1^) and lower D-band (1362 cm^−1^), further indicating that P,N-doped cellulose could promote the formation of char with a multilamellar carbonaceous structure.

### 3.6. Mechanism for Flame Retardancy

Based on the foregoing analysis, a reasonable mechanism relative to the condensed phase can be proposed to achieve excellent flame retardancy for HDPE composites with P,N-doped cellulose fibrils. The phosphoric acid hydrolysis of cellulose not only promotes the liberation of aggregated cellulose fibrils, but also introduces phosphate groups and ammonium phosphates into the surface of cellulose fibrils, forming P,N-doped cellulose fibrils. On the one hand, phosphoric acid hydrolysis causes the fibrils to be well dispersed in the HDPE matrix and promotes a compact char structure. By constructing a P,N-doped coatings on the surface of cellulose fiber, with increasing the temperature, the P,N-doped coatings at the surface of cellulose fibrils began to decompose, accompanied by the release of phosphoric acid which contributed to the dehydration reaction of cellulose fibrils, and then a crosslinking layer was formed at the surface of cellulose fibrils. Under a sealing space, the charcoal was formed due to the incomplete combustion of cellulose fibrils, leading to an intergrowth charring between cellulose fibrils and P,N-doped coatings. Therefore, the wick effect of cellulose was restrained. Compared with untreated cellulose, the P,N-doped cellulose decomposes during heating and induces the formation of an aromatic carbonaceous structure at a lower temperature. On the other hand, the char, which is a carbonaceous structure with a higher crystalline orientation, creates a physical barrier that blocks heat and oxygen from the flammable surface. Moreover, the formation of ammonium phosphates within the surface of P,N-doped cellulose decomposes to generate NH_3_. It can dilute the O_2_ concentration of the combustion zone, thus enhancing the flame retardancy of HDPE composites. The possible mechanism of flame retardancy is illustrated in Scheme 2.

## 4. Conclusions

Cellulose fibrils are prepared using phosphoric acid and aqueous ammonia in a one-pot reaction, which not only promotes the liberation of aggregated fibrils but also introduces elements P and N onto the cellulose fibril surface. During combustion, P,N-doped cellulose decomposes to produce pyro/polyphosphoric acid, which catalyzes the dehydration and esterification of cellulose. Moreover, it induces the formation of an aromatic carbonaceous structure at a lower temperature. The compact char structure with a higher crystalline orientation creates a physical barrier, thus making the flame retardancy of HDPE composites effective. In particular, HDPE composites with 7 wt % P,N-doped cellulose reached an optimum flame retardancy that corresponds to 25.7% LOI and 185.76 kW/m^2^ average HRR.

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
