# Peer review of "Flame Retardancy of High-Density Polyethylene Composites with P,N-Doped Cellulose Fibrils"

_polymers, 2020, doi:10.3390/polym12020336_

Round 1
Reviewer 1 Report
This work has prepared the P, N-doped cellulose fibrils using phosphoric acid and aqueous ammonia in one-pot reaction. The obtained P, N-doped cellulose fibrils with high liberation are thereafter incorporated to a HDPE matrix to improve the flame retardancy of HDPE composites. The results indicated that the composites have a significant improvement on flame retardancy of HDPE composites. This work would provide some insights for flame retardancy of HDPE composites. In general, the work is well-done and the manuscript is well-prepared. However, there are still some minor issues to be addressed.
Please explain why the addition of P, N-doped cellulose gives rise to increased smoke during burning. Please give flame retardancy rating according to the UL-94 standard. What about the mechanical properties of the HDPE composites with P, N-doped cellulose? How about the effect of higher amount of fillers on the properties of the composites? The manuscript should be carefully checked with language-proof. There are still some typos and grammar issues to be corrected. More explanations should be provided for the mechanism of flame retardancy. In introduction part, more background should be provided to show why authors use both P and N for doping.
Author Response
Responses to reviewer’s comments
Reviewer#1:
Point 1: Please explain why the addition of P, N-doped cellulose gives rise to increased smoke during burning.
Response 1: Thanks for your recommendation. Reasons for increased smoke of specimen are listed below: NFRPCs is often treated with phosphorus flame retardant (APP, red phosphorus), along with the improvement of flame retardancy, phosphorus flame retardant brings the problem of increased smoke release, especially in NFRPCs [1,2,3]. During the combustion process, phosphorus flame retardants, which act by flame poisoning/gas phase activity, interfere with the typical combustion radical reactions in the flame and result in incomplete combustion and thus brings a higher fire proof level together with a higher TSR index than untreated polymer.
In this paper, for pure HDPE specimen, it shows a low LOI index of 19.7 and could complete combustion and thus shows a low TSR of 1621.64 m2/m2.After the addition of cellulose or P, N-doped cellulose, time to form effective carbon layer is shortened and the density of carbon layer is enhanced, which leads to incomplete combustion of specimen and increased the TSR rate. What’s more, during the combustion process, P, N-doped cellulose would decompose to give out NH3 gas, which further contribute to the TSR increasement of specimen. It is our goal to pursue synchronous flame retardant and smoke suppression (i.e. both HRR and TSR are low), we will put more efforts in the future.
Seefeldt, H.; Braun, U.; Wagner, M.H. Residue stabilization in the fire retardancy of wood-plastic composites: combination of ammonium polyphosphate, expandable graphite, and red phosphorus. Macromol. Chem. Phys. 2012, 213, 2370-2377. Jiang, D.; Pan, M.; Cai, X.; Zhao, Y. Flame retardancy of rice straw-polyethylene composites affected by in situ polymerization of ammonium polyphosphate/silica. Compos. Part. A-Appl S. 2018, 109, 1-9. Nikolaeva, M.; Kärki, T. Reaction-to-fire properties of wood-polypropylene composites containing different fire retardants. Fire. Technol. 2015, 51(1), 53-65.
Point 2:Please give flame retardancy rating according to the UL-94 standard.
Response 2: Thanks for your recommendation. We investigated the flame retardancy using cone calorimetry and limited oxygen index (LOI) technologies in this paper. UL-94 test is also an important method for flame retardant properties. While for our system, HDPE composites with 1-7wt% P,N-doped cellulose have LOI values ranged from 20.4-25.7%, and can’t pass UL-94 test. For the flame retardant properties of HDPE composites with loading of P,N-doped cellulose, it will be also conduct with cone calorimetry, LOI and UL-94 test in the future.
Point 3:What about the mechanical properties of the HDPE composites with P, N-doped cellulose?
Response 3: Thanks for your recommendation. This part will be supplement in this paper as follows: Table 1 summarizes the mechanical properties of HDPE composites with untreated and P,N-doped cellulose fibrils.
Tensile property is one of most important mechanical properties for HDPE composites. HDPE with untreated cellulose has a tensile strength of 22.19 MPa and showed a 4.1% decrease compared to pure HDPE. For HDPE with P,N-doped cellulose fibrils, tensile strength first decreased and then increased with the addition of P,N-doped cellulose form 1 wt% to 9 wt%. HDPE with 9 wt% P,N-doped cellulose showed a tensile strength of 23.00 MPa, basically the same level of pure HDPE, which should be ascribed to the strengthen effects of cellulose fibrils.
On the other hand, toughness properties of HDPE composites were also summarized in Table 2. Pure HDPE showed a high elongation at break of 38.75 %. Addition of untreated cellulose or P,N-doped cellulose fibrils would significantly lower the elongation at break of HDPE. Impact Strength of HDPE showed the same trend. HDPE with 7 wt% P,N-doped cellulose showed the lowest impact strength of 9.38 kJ/m2, which showed a 84% decrease as compared to pure HDPE. Reduction of toughness of HDPE composites was ascribed to the poor interface compacity between HDPE matrix and cellulose fibrils. For HDPE with P,N-doped cellulose fibrils, the phenomenon was worsen for the inorganic P,N-doped coatings on cellulose fibril surface, which further decreased the toughness of HDPE composites.
Table 2. Mechanical properties of HDPE composites with P,N-doped cellulose.
|
Samples |
Tensile Strength |
Elongation at break |
Impact Strength |
|
MPa |
% |
kJ/m2 |
|
|
HDPE |
23.14 (1.28) |
38.75 (7.27) |
58.38 (8.63) |
|
untreated 7 wt% |
22.19 (0.94) |
34.91 (6.65) |
46.88 (14.76) |
|
P,N-doped 1 wt% |
21.57 (1.13) |
36.22 (8.75) |
58.44 (7.03) |
|
P,N-doped 3 wt% |
19.12 (1.04) |
29.96 (4.14) |
39.22 (9.16) |
|
P,N-doped 5 wt% |
19.32 (0.84) |
23.09 (3.45) |
13.28 (1.38) |
|
P,N-doped 7 wt% |
19.68 (3.16) |
22.50 (2.90) |
9.38 (0.77) |
|
P,N-doped 9 wt% |
23.00 (0.30) |
25.38 (4.79) |
13.592.20) |
Point 4 How about the effect of higher amount of fillers on the properties of the composites?
Response 4 : Thanks for your recommendation. In this paper, P, N-doped cellulose is obtained from rice straw following a complicated process. Rice straw is first extracted with benzene-methanol azeotrope, then delignified with sodium chlorite/acetic acid solution and stirred in NaOH solutions to remove of hemicellulose and finally a purified rice straw cellulose fiber is maintained. After that, purified cellulose fiber is alternatively treated with phosphonic acid and aqueous ammonia solutions to get a P, N-doped cellulose. When the P, N-doped cellulose adding amount reaches 9%, the flame retardancy (LOI and cone test) shows a significant improvement. When further increase the amount of fillers in HDPE composites, plenty of P, N-doped cellulose is needed that not economical acceptable. Our future work will study a higher amount of P, N-doped cellulose on the properties of HDPE composites.
Point 5: The manuscript should be carefully checked with language-proof. There are still some typos and grammar issues to be corrected. More explanations should be provided for the mechanism of flame retardancy.
Response 5: Thanks for your recommendation. This part will be supplement in this paper as follows: By constructing a P,N-doped coatings on the surface of cellulose fiber, with increasing the temperature, the P,N-doped coatings at the surface of cellulose fibrils began to decompose, accompanied by the release of phosphoric acid which contributed to the dehydration reaction of cellulose fibrils, and then a crosslinking layer was formed at the surface of cellulose fibrils. Under a sealing space, the charcoal was formed due to the incomplete combustion of cellulose fibrils, leading to an intergrowth charring between cellulose fibrils and P,N-doped coatings. Therefore, the wick effect of cellulose was restrained.
Point 6: In introduction part, more background should be provided to show why authors use both P and N for doping.
Response 6: Thanks for your recommendation. Additional description will be supplement in this background as follows:
Flame retardants containing P and N elements like chitosan, phytic acid, APP, DOPO are the most commonly appeared elements in environment friendly flame retardants. Gaan et al[4]. studied the effects of three nitrogen additives on the flame retardant action of cotton cellulose treated with tributyl phosphate (TBP). Results proved the synergistic effect between phosphorus and nitrogen that brings a better flame retardancy. Shi et al[5]. alternatively deposited PEI and APP on carbon fibers for flame retardancy of epoxy resin. With only 6 bilayers, a high flame retardancy (LOI of 41.0% and UL-94 V0 rating) was achieved. Wang et al. constructed a phosphorus-nitrogen containing polymer wrapped carbon nanotubes (CNT-PD) to improve the flame retardancy of epoxy resins. With 4 wt% addition of CNT-PD, LOI value of epoxy resins reached to 33.6%. In our experiment, phosphoric acid and aqueous ammonia were chosen to fabricate a P,N-doped coatings mainly for their economic acceptablity.
Gaan, S.; Sun, G.; Hutches K , Engelhard, M.H. Effect of nitrogen additives on flame retardant action of tributyl phosphate: phosphorus–nitrogen synergism. Polym. Degrad. Stabi. 2008, 93(1), 99-108. Shi, X.H.; Xu, Y.J.; Long, J.W.; Zhao, Q.; Ding, X.M.; Chen, L.; Wang, Y.Z. Layer-by-layer assembled flame-retardant architecture toward high-performance carbon fiber composite. Chem. Eng. J. 2018, 353, 550-558. Wang, S.; Xin, F.; Chen, Y.; Qian, L.; Chen, Y. Phosphorus-nitrogen containing polymer wrapped carbon nanotubes and their flame-retardant effect on epoxy resin. Polym. Degrad. Stabi. 2016, 129, 133-141.
Reviewer 2 Report
This paper concerns the flammability of HDPE/P-N modified cellulose fibrils composites. A huge effort of characterization of the system and of its flammability has been done and I think that this paper deserves to be published but some improvements are required.
For example, introduction is limited to the application of cellulose fibrils as Fire Retardant (FR) in some polymer (PLA and PU notably); however, nothing is reported on fire retardance of polyethylene with similar systems. That would enable the reader to recognize the merit of using fibrils instead of other polyhydric compounds in intumescent formulations.
The experimental part can be improved:
Line 86: “the rice straw granules were consecutively treated by extraction and delignification.” Please specify extraction solvent and delignification reactant.
Which apparatus was use to grinder HDPE pellets?
Did you notice initial charring during melt-compounding for 5-10 min at 170 °C?.
FTIR were surface ATR spectra or KBr disk?
How elemental analysis was carried out (cfr fig 2: C%, O%, P%, N%)?
Results: please delete lines 145-147
Scheme 1: why NH3 is considered to react with H3PO4 excess and not with grafted phosphate as well?
Flammability:
Table 1 nearly all parameters are in favor of a better fire retardance behavior of P-N doped fibrils composites but the SEA, TSR are quite worse and this could be a problem for applications, please comment on this. In addition, rate of weight loss MLR do no greatly differ in composites and in HDPE, this means that the rate of volatile evolution and fuel supply to the flame does not change very much. Comment on this also.
Additional comments: I do not like tridimensional representations in fig 4: why color legend is different in the three graphs? It is quite confusing. In addition CO, CO2 are not recognizable in the spectra.
Typing mistakes line 140 to investigate (and not investigated); line 183 amine group (and not mine groups)
Author Response
Responses to reviewer’s comments
Reviewer#2:
Reviewer #2: This paper concerns the flammability of HDPE/P-N modified cellulose fibrils composites. A huge effort of characterization of the system and of its flammability has been done and I think that this paper deserves to be published but some improvements are required.
Point 1: Introduction is limited to the application of cellulose fibrils as Fire Retardant (FR) in some polymer (PLA and PU notably); however, nothing is reported on fire retardance of polyethylene with similar systems. That would enable the reader to recognize the merit of using fibrils instead of other polyhydric compounds in intumescent formulations.
Response 1: Thanks for your recommendation. Additional description will be supplement in this background as follows:
Considering the successful application of treated cellulose fibrils as synergistic reinforcement phase and flame retardant in polymers like PLA and PU, its application in one of the most widely used synthetic plastics-polyethylene is still lack of investigation. Though natural cellulose reinforced polyethylene has been widely studied [1,2,3], flame retardant treatment of such materials still followed a bulk mode that directly adding FRs into the polyethylene matrix.
Zhang, F.; Qiu, W.; Yang, L.; Endo, T.; Hirotsu, T. Mechanochemical preparation and properties of a cellulose–polyethylene composite. J. Materials .Chem. 2002, 12(1), 24-26. Araujo, J.R.; Mano, B.; Teixeira, G.M.; Spinace, M.A.S.; De Paolia, Marco-A. Biomicrofibrilar composites of high density polyethylene reinforced with curauá fibers: mechanical, interfacial and morphological properties. Compos. Sci. Technol. 2010, 70(11), 1637-1644. Diallo, A.K.; Jahier, C.; Drolet, R.; Tolnai, B.; Montplaisir, D. Cellulose filaments reinforced low‐density polyethylene. Polym. Composite. 2019, 40, 16-23.
Point 2: Line 86: “the rice straw granules were consecutively treated by extraction and delignification.” Please specify extraction solvent and delignification reactant.
Response 2: Thanks for your recommendation. The rice straw was first extracted with Benzyl-alcohol mixtures and then delignated following with acetic acid and sodium chlorite.
Point 3: Which apparatus was use to grinder HDPE pellets?
Response 3: Thanks for your recommendation. A plastic mill (NF-200, Kim Jung-Teaching Instrument Co., LTD, China) was used to grinder HDPE pellets in our experiment.
Point 4: Did you notice initial charring during melt-compounding for 5-10 min at 170 °C?
Response 4: Thanks for your recommendation. During the melt-compounding process, we noticed a slightly color bum of HDPE composite.
Point 5: FTIR were surface ATR spectra or KBr disk?
Response: Thanks for your recommendation. FTIR were conducted with ATR spectra.
Point 6: How elemental analysis was carried out (cfr fig 2: C%, O%, P%, N%)?
Response: Thanks for your recommendation. Elemental content of C, O, P, N were carried out with an energy dispersive spectroscopy (EDS, OXFORD Instrument) instrument together with SEM test.
Point 7: Please delete lines 145-147.
Response: Thanks for your recommendation. Accepted.
Point 8: Why NH3 is considered to react with H3PO4 excess and not with grafted phosphate as well?
Response 8: Thanks for your recommendation. Accepted. NH3 should also react with grafted phosphate as well, relative description would be revised.
Point 9: Table 1 nearly all parameters are in favor of a better fire retardance behavior of P-N doped fibrils composites but the SEA, TSR are quite worse and this could be a problem for applications, please comment on this. In addition, rate of weight loss MLR do no greatly differ in composites and in HDPE, this means that the rate of volatile evolution and fuel supply to the flame does not change very much. Comment on this also.
Response 9: Thanks for your recommendation. NFRPCs is often treated with phosphorus flame retardant (APP, red phosphorus), along with the improvement of flame retardancy, phosphorus flame retardant brings the problem of increased smoke release, especially in NFRPCs [1,2,3]. During the combustion process, phosphorus flame retardants, which act by flame poisoning/gas phase activity, interfere with the typical combustion radical reactions in the flame and result in incomplete combustion and thus brings a higher fire proof level together with a higher TSR index than untreated polymer.
In this paper, for pure HDPE specimen, it shows a low LOI index of 19.7 and could complete combustion and thus shows a low TSR of 1621.64 m2/m2.After the addition of cellulose or P, N-doped cellulose, time to form effective carbon layer is shortened and the density of carbon layer is enhanced, which leads to incomplete combustion of specimen and increased the TSR rate. What’s more, during the combustion process, P, N-doped cellulose would decompose to give out NH3 gas, which further contribute to the TSR increasement of specimen. It is our goal to pursue synchronous flame retardant and smoke suppression (i.e. both HRR and TSR are low), we will put more efforts in the future.
For MLR data, it is observed that no greatly differ in composites and in HDPE. However, there is no dominated relevance between MLR and flame retardancy of polymers. Specific reasons for MLR data should be ascribed to the characteristic of P, N-doped cellulose that we will study in our future work.
Seefeldt, H.; Braun, U.; Wagner, M.H. Residue stabilization in the fire retardancy of wood-plastic composites: combination of ammonium polyphosphate, expandable graphite, and red phosphorus. Macromol. Chem. Phys. 2012, 213, 2370-2377. Jiang, D.; Pan, M.; Cai, X.; Zhao, Y. Flame retardancy of rice straw-polyethylene composites affected by in situ polymerization of ammonium polyphosphate/silica. Compos. Part. A-Appl S. 2018, 109, 1-9. Nikolaeva, M.; Kärki, T. Reaction-to-fire properties of wood-polypropylene composites containing different fire retardants. Fire. Technol. 2015, 51(1), 53-65.
Point10: I do not like tridimensional representations in fig 4: why color legend is different in the three graphs? It is quite confusing.
Response 10: Thanks for your recommendation. Color legend in Fig.4 differs from each is meant to show the total curves in a suitable area in the figure.
Point 11: Typing mistakes line 140 to investigate (and not investigated); line 183 amine group (and not mine groups)
Response: Thanks for your recommendation. Accepted.
